# Photochemistry of the pyruvate anion produces $CO_2$, CO, $CH_3^-$, $CH_3$, and a low energy electron

Connor J. Clarke [1], Jemma A. Gibbard [1], Lewis Hutton[1], Jan R. R. Verlet [1✉] & Basile F. E. Curchod [1✉]

The photochemistry of pyruvic acid has attracted much scientific interest because it is believed to play critical roles in atmospheric chemistry. However, under most atmospherically relevant conditions, pyruvic acid deprotonates to form its conjugate base, the photochemistry of which is essentially unknown. Here, we present a detailed study of the photochemistry of the isolated pyruvate anion and uncover that it is extremely rich. Using photoelectron imaging and computational chemistry, we show that photoexcitation by UVA light leads to the formation of $CO_2$, CO, and $CH_3^-$. The observation of the unusual methide anion formation and its subsequent decomposition into methyl radical and a free electron may hold important consequences for atmospheric chemistry. From a mechanistic perspective, the initial decarboxylation of pyruvate necessarily differs from that in pyruvic acid, due to the missing proton in the anion.

[1] Department of Chemistry, Durham University, Durham DH1 3LE, United Kingdom. ✉email: j.r.r.verlet@durham.ac.uk; basile.f.curchod@durham.ac.uk

Pyruvic acid and its conjugate base, the pyruvate anion, $CH_3COCOO^-$ (Fig. 1), are pervasive throughout nature. Both are present within seawater, fogs, aerosols, clouds, and the atmosphere[1,2], having biogenic and anthropogenic origins[3–6]. Their presence in atmospheric aerosols has attracted particular attention due to the rich photochemistry of pyruvic acid, which differs between the gas- and solution phase and at their dividing interface[7–9], and because pyruvic acid serves as a representative α-dicarbonyl in atmospheric models[10]. The primary photo-dissociation process upon UVA excitation is decarboxylation (>97%) to form short-lived methylhydroxycarbene that rearranges to acetaldehyde[11–14]. The decarboxylation process is facilitated by an intramolecular proton transfer in the $S_1$ excited state, with additional possible involvement of triplet states depending on the environment[11,15]. The photoproducts can go on to form a series of other species: in aqueous environments, these include acetoin, lactic acid, acetic acid, and oligomers[10]. It is through such reactions that pyruvic acid has been considered a precursor for primitive metabolism[16]. However, while the photochemistry of pyruvic acid has been extensively studied, that of the pyruvate anion has not, despite its necessary prevalence in aerosols and seawater as the conjugate base of pyruvic acid[17,18]. There are interesting questions regarding its intrinsic photochemistry as pyruvate lacks the proton involved in the intramolecular proton transfer mechanism of pyruvic acid—a necessary step in pyruvic acid prior to decarboxylation. Here, we present a combined photoelectron spectroscopic and computational study of the isolated pyruvate anion, revealing the rich photochemistry of pyruvate that can be initiated by UVA radiation. We demonstrate that the pyruvate anion not only experiences decarboxylation, but also a subsequent unimolecular decay to form CO and a methide anion, $CH_3^-$. The methide anion further decomposes into $CH_3$ and a free electron.

## Results and discussion

Gas-phase pyruvate anions were produced by electrospray ionisation and mass-selected before being excited at a range of photon energies, $h\nu$, and the subsequently detached electrons were monitored using an imaging photoelectron spectrometer. Figure 2(a) presents the photoelectron spectra acquired with photon energies close to the actinic region, between $3.3 \leq h\nu \leq 4.3$ eV, in increments of 0.1 eV. As each spectrum has been normalised to its maximum signal, comparison of photoelectron intensities across different photon energies should be done with caution and are not representative of excitation or detachment cross-sections. Two clear features are present in the recorded photoelectron spectra (Fig. 2(a)): a broad peak centred near an electron kinetic energy (eKE) of 0.5 eV for $h\nu = 4.3$ eV, which red-shifts with decreasing $h\nu$; and a sharp feature present at all

photon energies, but especially for $3.3 \leq h\nu \leq 3.9$ eV, peaking at eKE = 0 eV.

The broad photoelectron peak in the $3.8 \leq h\nu \leq 4.3$ eV spectra, which red-shifts with decreasing $h\nu$, can be attributed to direct electron detachment of the pyruvate anion, forming the neutral $D_0$ ground state. To support this assignment, we calculated the photoelectron spectra of the pyruvate anion using the nuclear ensemble approach (NEA) with (U)DFT/ωB97X-D/aug-cc-pVDZ (see discussion below and further information on the calculations in the Methods section). The experimental $h\nu = 4.3$ eV spectrum (displayed as a function of electron binding energy, eBE = $h\nu$ – eKE) and the calculated photodetachment spectrum for the pyruvate anion are in excellent overall agreement (Fig. 3(a)), with disparity at high eBE stemming from the low eKE electron loss channel, which is described below. The maximum of the spectrum corresponds to the experimental vertical detachment energy of $3.8 \pm 0.1$ eV. The onset of the photoelectron signal occurs at ~3.3 eV, which is comparable to the electron affinities of many other carboxylates[19–21]. The calculated photoelectron spectrum accounts for the vibrational distribution of pyruvate in its ground

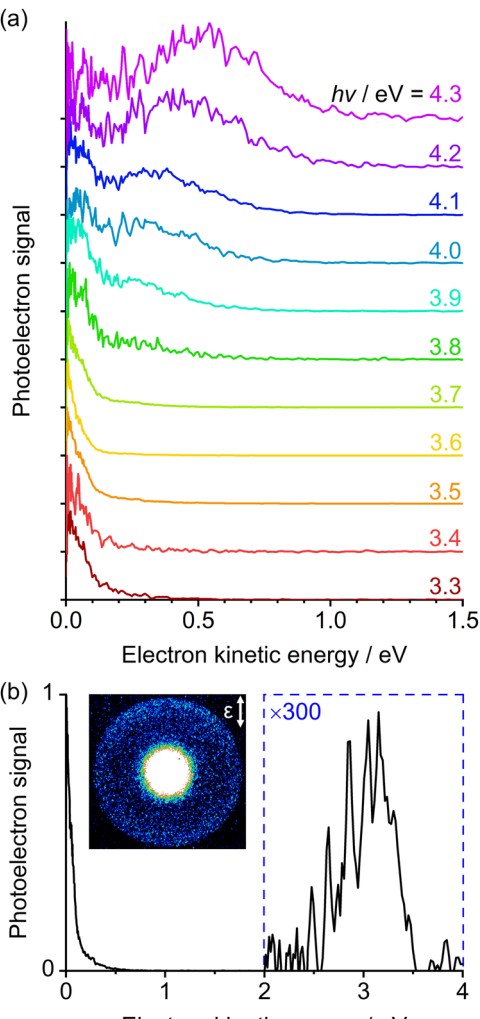

**Fig. 2 Photoelectron spectra following irradiation of the pyruvate anion using ns laser pulses with photon energy $h\nu$.** (**a**) Spectra at a range of $h\nu$, normalised to their maximum intensity and offset for clarity. (**b**) Spectrum at $h\nu = 3.5$ eV in which the signal at eKE > 2 eV has been amplified to highlight a feature at high energy. Inset is a photoelectron image in which the centre has been saturated to highlight this high energy (large radius) feature. The polarisation of the light, $\varepsilon$, is indicated by the double arrow.

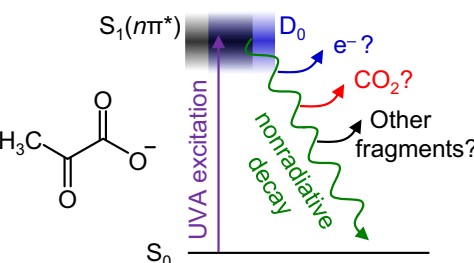

**Fig. 1 Photodynamics of pyruvate anion.** Chemical structure of pyruvate and schematic of possible photochemical pathways initiated by UVA excitation e.g., electron emission, $CO_2$ release, formation of other molecular fragments.

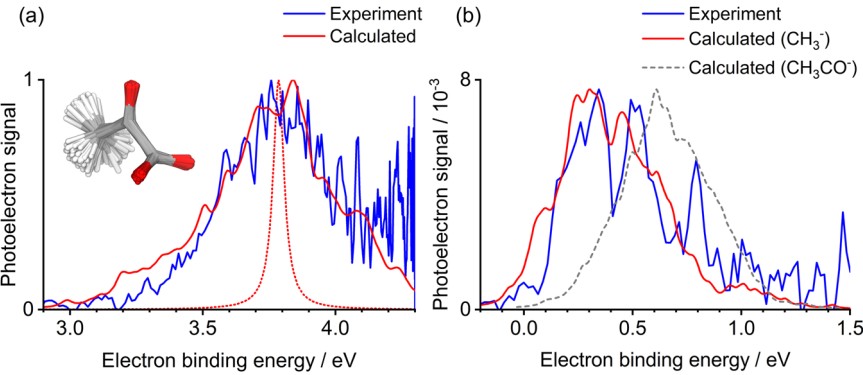

**Fig. 3 Experimental and theoretical photoelectron spectra.** Measured photoelectron spectra in terms of eBE = $h\nu$ – eKE for (**a**) $h\nu$ = 4.3 eV and (**b**) $h\nu$ = 3.5 eV are shown in blue. $D_0 \leftarrow S_0$ calculated photoelectron spectra for (**a**) pyruvate and (**b**) the methide anion, obtained with the nuclear ensemble approach (NEA) using (U)DFT/ωB97X-D/aug-cc-pVDZ, are given in red. A similarly calculated spectrum for the acetyl anion is presented in (**b**) as a grey dashed line. The red dotted line in (**a**) shows the contribution of the optimised ground state geometry to the calculated spectrum. Inset in (**a**) is an overlap of geometries used to generate the pyruvate photoelectron spectra.

vibrational state. To achieve this goal, the NEA calculates a Wigner distribution (within a harmonic approximation) for the ground vibrational state of pyruvate from which a certain number of geometries can be sampled (500 in this case, depicted in the inset of Fig. 3(a))[22]. The influence of properly accounting for the vibrational distribution of pyruvate is clearly illustrated by comparing the NEA photoelectron spectrum with the broadened $D_0 \leftarrow S_0$ vertical energy calculated with the same level of theory, but only at the ground-state optimised geometry (red dotted curve in Fig. 3(a)). We note that the ground-state structure of pyruvate is significantly different from that of pyruvic acid, as the carboxylate group in the former is perpendicular to the plane of the molecule (the planar structure of pyruvate is a transition state)[23–25]. Finally, we also measured the photoelectron angular distributions, quantified by an anisotropy parameter $\beta_2$. This lies between +2 and −1 to describe a $\cos^2\theta$ and $\sin^2\theta$ distribution, respectively, where $\theta$ is the angle between the outgoing electron and the polarisation axis of the incident light[26]. For the direct detachment feature at $h\nu$ = 4.3 eV, $\beta_2 \approx -0.2$, which is consistent with the predominantly non-bonding p-orbital nature of the orbital from which the electron is removed (see Supplementary Figure 1)[27].

The opening of the direct detachment channel of pyruvate lies in the UVA spectral region where both pyruvic acid and pyruvate absorb. To the best of our knowledge, the absorption spectrum for the isolated pyruvate anion has yet to be measured. However, given the similarity of the absorption spectra of the acid and its conjugate base in solution and the modest solvatochromatic shifts seen in the acid, we anticipate that the $S_1 \leftarrow S_0$ absorption of the isolated anion peaks in a similar range to that of the acid, around $h\nu$ ~3.5 eV[13]. This expectation is in broad agreement with that predicted by Shemesh et al.[28]. The calculated photoabsorption cross-section for the $n\pi^*$ transition of pyruvate using the NEA and SCS-ADC(2)/cc-pVTZ (Supplementary Fig. 2) exhibits a maximum at 3.9 eV, with significant intensity near 3.5 eV. Hence, upon UVA excitation to the $S_1$ state, electrons can be lost directly. Alternatively, electrons can detach through indirect processes in which the anion first undergoes nonradiative decay (Fig. 1) before either losing an electron or absorbing a second photon in the ~5 ns duration laser pulse.

In Fig. 2(a), we observe a feature peaking at eKE = 0 eV across a broad range of photon energies, with a spectral shape that represents a Boltzmann distribution. Such signals are characteristic of statistical (or thermionic) electron emission, typically from the ground electronic state of an anion, where the total internal energy after photoabsorption exceeds the binding energy of the

electron. In competition with statistical electron loss, unimolecular decay by dissociation is possible. For pyruvate, we expect the latter to dominate because the electron binding energy is larger than the typical bond energy associated with $CO_2$ loss[29]. In this case, an anionic fragment with a lower electron binding energy may form and this fragment anion may also lose electrons by thermionic emission. Some low-energy electron emission may also arise from excited states by fast auto detachment that competes with internal conversion or dissociation. But, regardless of the exact mechanism that leads to the feature peaking at eKE = 0 eV, its presence necessitates the absorption of a photon and some formation of a ground state anion. Enhanced thermionic emission was observed in the $h\nu$ = 3.5 – 3.7 eV spectra, evidenced in Fig. 2(a) by the enhanced signal-to-noise ratio in these photoelectron spectra, suggesting an excited state is located ~ 3.6 eV above the ground state, consistent with the absorption spectrum of gas-phase pyruvic acid[30].

In addition to the two clear electron emission features, a very weak feature at high eKE (low binding energy) appeared in all acquired photoelectron images up to $h\nu$ = 4.1 eV, but with the greatest intensity in the $h\nu$ = 3.5 – 3.6 eV range. An example spectrum is shown in Fig. 2(b) at $h\nu$ = 3.5 eV, demonstrating that this high kinetic energy signal has <1% the intensity of the corresponding thermionic emission feature. In Fig. 3(b), this feature is replotted in terms of electron binding energy. The energy at which this broad feature peaks does not correlate with a two-photon resonance-enhanced detachment process, nor with a sequential multiple-photon process from the pyruvate anion. Instead, we assign the feature to a photofragment of pyruvate. That is to say, dissociation is induced by an initial photon absorbed by the pyruvate anion, and then a second photon detaches an electron from the resulting anion fragment. We first considered the possibility of decarboxylation, as observed in the photolysis of pyruvic acid. This produces the acetyl anion, $CH_3CO^-$. Comparing the weak feature with a previous photoelectron spectroscopy measurement of the acetyl anion (see Supplementary Figure 3)[31] or its calculated photoelectron spectrum, as shown in Fig. 3(b), shows that the acetyl anion is *not* the observed fragment. In particular, the adiabatic electron affinity of $CH_3CO$ is ~0.4 eV, in excess of the adiabatic electron affinity of ~0.1 eV for our observed fragment.

If the fragment does not appear to be formed by $CO_2$ loss, then what else could it be? Using an exploratory ab initio molecular dynamics (AIMD) simulation to investigate the behaviour of the acetyl anion in its ground electronic state revealed that the methide anion, $CH_3^-$, could be formed. Specifically, during the AIMD

conducted in the NVE ensemble, the ground-state acetyl anion appeared to be unstable with respect to CO loss already at an average temperature of 1400 K. Based on this observation, we calculated the photoelectron spectrum of $CH_3^-$ and compared it with the experimental spectrum at $hv = 3.5$ eV (Fig. 3(b)). The agreement is excellent and the overall spectrum is in accordance with high-resolution photoelectron imaging measurements by Lineberger and coworkers (see Supplementary Figure 3)[32]. Furthermore, the anisotropy of the feature taken from the photoelectron image inset in Fig. 2(b) was determined to be $\beta_2 \approx +0.4$, which agrees well with their modified Wigner-Bethe-Cooper-Zare equation for $CH_3^-$ photodetachment[32,33].

Based on the analysis above, we propose that the following overall photochemical processes are present:

$$CH_3COCOO^- + hv_{UVA} \rightarrow CH_3CO^- + CO_2,$$
$$CH_3CO^- \rightarrow CH_3^- + CO, \quad\quad (1)$$
$$CH_3^- \rightarrow CH_3 + e^-$$

In our experiment, we do not observe the acetyl anion. This is likely because $CH_3CO^-$ is formed with sufficient internal energy to lose CO and is therefore expected to act as a short-lived intermediate (<ns) en route to forming the methide anion. Thence, we also do not expect $CH_3CO^-$ to contribute significantly to the thermionic emission signal observed in Fig. 2. Instead, the final product $CH_3^-$ has no further energetically accessible dissociation routes but does have a very low adiabatic electron detachment energy. Therefore, one may expect that a significant fraction of the $CH_3^-$ goes on to form methyl radicals through thermionic emission. As thermionic emission typically proceeds on a timescale exceeding many nanoseconds, $CH_3^-$ may absorb a second photon in the 5 ns laser pulse to lead to its observation in the photoelectron spectrum (Fig. 3(b)). The photon flux dependence of the $CH_3^-$ peak is considered in Supplementary Note 1.

While our combined experimental and computational study has clearly identified key products in the photochemistry of pyruvate, it does not provide details on the product yields, nor on the detailed mechanisms. However, we noted that the observation of $CH_3^-$ requires a sequential two-photon process, which typically has a low probability, and thus the small-signal associated with the methide anion should not be taken as an indicator of low quantum yield. Moreover, if the thermionic emission feature arises predominantly from $CH_3^-$, as we suspect, then the sequential fragmentation appears to be the dominant decay pathway for the UVA-excited pyruvate anion, and it must occur within a few nanoseconds. A potential competing pathway, to which our current experiment is blind, is $CO_2$ loss from the neutral potential energy surface of the pyruvate anion (i.e. dissociative photodetachment)[34]. Indeed, our calculations suggest that the pyruvate neutral surface is dissociative with respect to $CO_2$ (which also motivated the use of the NEA to modelling the photoelectron spectrum). However, dissociative photodetachment requires access to the direct detachment continuum, which becomes only prominent for $hv > 3.8$ eV or the UVB spectral range (see Fig. 2). Extensive ground-state dynamics following photoexcitation and subsequent decarboxylation, as observed here for the pyruvate anion, has also been observed in a range of other carboxylate anions such as *p*-coumarate[35] and octatrienoate[36]. Hence, whilst the dynamics observed in pyruvate are surprising, they are not unprecedented and highlight the importance of ground-state dynamics following photoabsorption, much of which is likely to be athermal[37,38].

From an atmospheric perspective, the photochemistry of pyruvate is pertinent due to its relative ubiquity in solution (at least compared to pyruvic acid) and ability to absorb sunlight (UVA),

even at sea level. The $pK_a$ of pyruvic acid is 2.5 and has been measured to be as low as 0.7 at the water/air interface[39]. Hence, virtually *all* pyruvic acid is deprotonated in seawater sprays and will dominate in all but the most acidic aerosols. Naturally, the vapour pressure of the anion is much lower than that of pyruvic acid and so, the role of the pyruvate anion as an isolated species may be less important than that of the acid. Nevertheless, the study of the intrinsic dynamics offers important insight into the photo-induced decay pathways that are operable. The photochemical production of the methide anion – the simplest of carbanions – raises questions on its possible reactivity with other atmospheric compounds, which has not been considered previously, except in the atmosphere of Titan[32,40]. The methide anion can also decay into $CH_3$ and a free electron. This low energy free (or partially solvated) electron can react with surrounding molecules[41,42]. The $CH_3$, which is normally formed from the reaction of methane with OH, reacts with $O_2$ to form methyl peroxide[43]. Hence, the previously overlooked pyruvate anion in gas phase has the ability to produce a range of exotic and reactive species in the atmosphere following photoabsorption. How micro-solvation affects the photochemistry of the pyruvate anion remains an open question. To this end, understanding the basic mechanism of pyruvate decarboxylation and subsequent methide production is a critical step that remains unknown as it necessarily follows different decay dynamics from the primary $CO_2$ loss pathway seen in pyruvic acid, due to the absent proton.

## Methods

**Experimental**. The experiment comprises a velocity-map imaging photoelectron spectrometer coupled to a time-of-flight mass spectrometer equipped with an electrospray ionisation source[44,45]. A ~100 mM solution of pyruvic acid (Sigma-Aldrich) in methanol underwent electrospray ionisation, producing the gaseous deprotonated anion, pyruvate. The anions were electrostatically trapped and thermalised to room temperature and then injected and accelerated in a Wiley-McLaren time-of-flight spectrometer for mass separation. Nanosecond pulses (~5 ns FWHM) from an Nd:YAG (Continuum Surelite) laser pumped an optical parametric oscillator and were subsequently loosely focussed in the interaction region where they intercepted a mass-selected ion packet containing the pyruvate anions. Emitted electrons were collected in a velocity-map imaging spectrometer[46] and analysed using polar onion peeling to offer the kinetic energies and angular distribution of the liberated photoelectrons[47]. The resolution of the spectrometer was ~5% of the eKE and was calibrated using the photoelectron spectrum of iodide and hydride.

**Computational Details**. The ground-state ($S_0$) optimised geometry and corresponding vibrational frequencies were obtained for pyruvate and methide anion using density functional theory (DFT) with the $\omega$B97X-D functional[48] and the aug-cc-pVDZ basis set[49]. For each molecule, a Wigner distribution for uncoupled harmonic oscillators was constructed from the DFT ground-state geometry and corresponding vibrational frequencies, from which 500 geometries were randomly sampled assuming that the molecule is in its ground vibrational state (0 K). For each geometry, the vertical ionisation energy to $D_0$ was calculated by a difference of electronic energy using DFT/$\omega$B97X-D/ aug-cc-pVDZ for $S_0$ and UDFT/$\omega$B97X-D/ aug-cc-pVDZ for $D_0$, and the corresponding intensity was approximated by using the norm of the Dyson orbital. The electron binding energy obtained with this level of theory for each molecule at its ground-state geometry (obtained with (U)DFT/ $\omega$B97X-D/aug-cc-pVDZ) is in excellent agreement with that obtained with (U)

**Table 1 Electron binding energy calculated with (U)DFT and (U)CCSD(T)-F12 for the optimised ground-state geometry of each molecule obtained with DFT/$\omega$B97X-D/aug-cc-pVDZ.**

| | electron binding energy (eV) | | |
| --- | --- | --- | --- |
| | Methide anion | Acetyl anion | Pyruvate anion |
| (U)DFT/$\omega$B97X-D aug-cc-pVDZ | 0.381 | 0.606 | 3.786 |
| (U)CCSD(T)-F12 aug-cc-pVTZ | 0.395 | 0.665 | 3.906 |

CCSD(T)-F12 (Table 1). The (U)CCSD(T)-F12 calculations were conducted with Molpro 2012.

The photoelectron signal for each geometry is obtained by placing a Lorentzian function centred at the vertical ionisation energy, with a height proportional to the Dyson norm, and a phenomenological width of 0.05 eV. The final photoelectron spectrum was calculated in a nuclear ensemble approach by averaging over all the signals provided by the 500 geometries (this strategy is coined the Dyson orbital norm approach)[22,50].

The NEA has also been employed for calculating the photoabsorption cross-section of pyruvate for the lowest $n\pi^*$ band. 500 geometries were sampled from the same Wigner distribution as used for the photoelectron spectrum of pyruvate. For each geometry selected, the vertical transition energy to $S_1$ and corresponding oscillator strength were calculated using SCS-ADC(2) and a cc-pVTZ basis set. The photoabsorption cross-section was obtained by averaging the transitions (broadened by a Lorentzian with a 0.05 eV width) over all the 500 geometries.

Exploratory ab initio molecular dynamics simulation of the acetyl anion were performed with the GPU-accelerated TeraChem v1.9 package using UDFT/$\omega$B97X-D/aug-cc-pVDZ for the electronic structure[51,52]. The AIMD simulation was initiated from the ground-state DFT optimised geometry with initial nuclear velocities sampled from a Boltzmann distribution at 2200 K, and conducted within a NVE ensemble with a time step of 0.5 fs. Redistribution of the initial kinetic energy into potential energy leads to an ab initio molecular dynamics with an average temperature of 1400 K. The dissociation energy ($D_e$) of acetyl anion leading to the formation of CO and $CH_3^-$ is calculated to be 113.3 kJ mol$^{-1}$ at the DFT/$\omega$B97X-D/aug-cc-pVDZ level of theory and 93.1 kJ mol$^{-1}$ at the CCSD(T)-F12/aug-cc-pVTZ (using the same DFT optimised structures).

The DFT calculations were performed with Gaussian09[53], while Turbomole 7.3.1 was used for SCS-ADC(2)[54]. The Newton-X 2.0 package[55] was used to calculate the Wigner distribution, the sampling, as well as the production of the photoelectron spectrum from the (U)DFT results.

## Data availability

The experimental and computational data generated in this study (raw photoelectron images, photoelectron spectra and calculated spectra) have been deposited and can be accessed at https://doi.org/10.5281/zenodo.5776603.

## Code availability

Polar onion peeling is available at https://www.github.com/adinatan/PolarOnionPeeling (Matlab version) or http://www.verlet.net/research.html (Labview version).

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

## Acknowledgements

Jemma A. Gibbard is grateful for support from a Ramsay Memorial Fellowship. Connor J. Clarke is thankful for the supporting Durham Doctoral Scholarship. This project has received funding from the European Research Council (ERC) under the European Union's Horizon 2020 research and innovation programme (Grant agreement No. 803718, project SINDAM). This article is based upon work from COST Action CA18212 — Molecular Dynamics in the GAS phase (MD-GAS), supported by COST (European Cooperation in Science and Technology), and made use of the facilities of the Hamilton HPC Service of Durham University.

## Author contributions

J.R.R.V. and B.F.E.C. conceived the project. C.J.C. and J.A.G. performed the experiments and L.H. the calculations with help and contributions from J.R.R.V. (experiment) and B.F.E.C. (theory). All analysed the data and discussed the results. C.J.C., J.R.R.V., and B.F.E.C. wrote the paper with input from L.H. and J.A.G.

## Competing interests

The authors declare no competing interests.
