## [Peer Review File · Nature Communications]

REVIEWER COMMENTS

Reviewer #1 (Remarks to the Author):

The manuscript presents a combined experimental and theoretical study of photodecomposition of the pyruvate anion, which is a conjugate base of pyruvic acid. The authors employed the photoelectron imaging technique and DFT molecular dynamics simulations to qualitatively determine the photodissociation products. While the work is interesting and technically solid in general, I don't see it as sufficiently important to make a cut for publication in the Nature group of journals. The authors point at potential importance for atmospheric chemistry mentioning the relative abundance of pyruvic acid in the sea water, mostly in the form of its conjugate base. However, they argue themselves that "the vapour pressure of the anion is much lower than that of pyruvic acid and so, the role of the pyruvate anion as an isolated species may be less important than that of the acid." The study does not provide any kinetic data required for atmospheric kinetic models and without kinetic modeling it is impossible to say whether the photodissociation of the pyruvate anion can make any significant contribution to the atmospheric processes. Moreover, the authors single out the formation of methyl radical calling it "a highly reactive species that can contribute to the formation of secondary organic aerosols". In fact, at atmospheric temperatures CH_3 can only rapidly recombine with other radicals but is virtually unreactive with closed shell species.

Using MD simulations with ab initio (DFT) potentials is fancy but the results are only as good as the underlying potentials. DFT methods do not generally provide chemical accuracy comparable with, e.g. coupled cluster theory. I wonder if the less than spectacular agreement of the measured and calculated photoelectron spectra (Fig. 3) is due to the deficiency of the DFT energetics. Also, in my opinion, more accurate estimates of the lifetime of the CH_3CO^- anion can be obtained using statistical (RRKM) calculations of its decomposition rate constant using more accurate (CCSD(T)-like) energies of the pertinent local minima and transition states, instead of using DFT-AIMD. In any case, validation of the chosen density functional vs. more accurate electronic structure methods is required before the results of the AIMD calculations can be trusted.

Reviewer #2 (Remarks to the Author):

Given the ubiquity of the pyruvic acid & pyruvate anion in solutions and at air/water interfaces, they play important roles in aerosol chemistry and atmospheric environments. Their photochemistry in actinic region is particularly relevant to their chemical transformations. Authors applied the state-of-the-art ion spectroscopy and advanced theoretical methods to carry out a comprehensive probe of the pyruvate anion in its isolated form. Authors excited the anion to its first excited state in the

actinic regime, and watched the decay and dissociation channels via VMI photoelectron spectroscopy. Channels such as direct electron detachment, resonant processes that led loss of CO, CO₂ and producing CH₃- (methide) were detected, providing a rich and complex photochemistry for this important anion. Sophisticated theories were applied to support experimental observations. Overall this is a well executed work. The unraveled rich photochemistry should have important implications to understand the pyruvic acid transformation in the atmospheric chemistry setting. The discovery of producing methide and methyl radical is particularly worth noting. I would strongly recommend its publication.

I only have a couple of technique questions:

what is the temperature used in simulating direct detachment channel for CH₃COCOO⁻ (Fig 3 a) via NEA approach?

The weak spectral signature for CH₃⁻ is proposed via two-photon process. Was there any photon-flux dependent study conducted?

Typo: top of page 4, Figure 1 - Figure 2

Reviewer #3 (Remarks to the Author):

The communication describes a joint experimental and theoretical study of the title molecule, providing new information on its possible photodissociation pathways. How pyruvate is decomposed plays a potential role in the cycles of atmospheric chemistry, and the fact that it may produce methide radicals is a novel, potentially important, observation. The paper also shows novelty in the good agreement between experiment and theory, supporting the role calculations have to play in unravelling the signals from time-resolved spectroscopy and demonstrating the accuracy of present state-of-the-art calculations.

Both of these novelties are of significance for the field of chemical dynamics, and the decomposition pathway of pyruvate of wider significance in physical and atmospheric chemistry. The paper is a sound and well argued piece of work, and as a result I support its publication in Nature Communications. I have a few minor points that the authors should consider.

1. on p4 top "Figure 1 presents..." should refer to Figure 2.

2. on p5 it is mentioned that the ground-state structure of pyruvate has a perpendicular carboxylate group, whereas pyruvic acid is planar. This is surprising as the electronic structure would not be expected to be very different so perhaps the authors could give a brief explanation as to why this is.

3. on p5 it is noted that the experiments show that the photoelectrons from pyruvate come from a non-bonding p-orbital. Does this tie in with the calculated electronic structure for the radical ground-state?

4. on p6 when discussing the "thermionic emission" signal, it is stated that this "... necessitates the absorption of a photon and the formation of a ground state anion". Does thermionic emission really require the absorption of a photon - it can occur if a molecule is heated to a high enough temperature? And does it really require that the electron is lost from the ground-state? Can it not occur for an electronic excited state as long as there is enough time for loss of coherence?

5. on p6 it is stated that "Enhanced thermionic emission was observed in the $h\nu = 3.5 - 3.7$ eV spectra,...". I do not see this in Figure 2. Furthermore, what is the relevance of the excited-state that may lie at 3.6eV? May this provide a resonant state that results in thermionic emission (which contradicts the statement discussed above in Q4). Or is this the pathway to fragmentation of the excited anion?

6. on p7 "a very weak feature at high eKE..." is mentioned and then discussed. This is not visible in the curves of Figure 2 and this should be made clear.

7. on p7 the various possible fragments are discussed and discarded. For completeness, what about the formation of the CO_2^- ion?

December 15, 2021

We thank the Reviewers for their positive and constructive comments on our manuscript "*Photochemistry of the pyruvate anion produces CO₂, CO, and CH₃*".

We have updated the manuscript to address all the Reviewers' suggestions. We detail these changes and include responses to the Reviewers' questions below. We reproduce below the entire Reviewer comments in **red** for clarity, and additions/corrections to the text are denoted in **blue**. We also uploaded an annotated version of the revised manuscript, where all modifications are indicated in **blue**.

Report of Reviewer #1

Reviewer #1 (Comments to the Author):

The manuscript presents a combined experimental and theoretical study of photodecomposition of the pyruvate anion, which is a conjugate base of pyruvic acid. The authors employed the photoelectron imaging technique and DFT molecular dynamics simulations to qualitatively determine the photodissociation products. While the work is interesting and technically solid in general, I don't see it as sufficiently important to make a cut for publication in the Nature group of journals. The authors point at potential importance for atmospheric chemistry mentioning the relative abundance of pyruvic acid in the sea water, mostly in the form of its conjugate base. However, they argue themselves that "the vapour pressure of the anion is much lower than that of pyruvic acid and so, the role of the pyruvate anion as an isolated species may be less important than that of the acid." The study does not provide any kinetic data required for atmospheric kinetic models and without kinetic modeling it is impossible to say whether the photodissociation of the pyruvate anion can make any significant contribution to the atmospheric processes. Moreover, the authors single out the formation of methyl radical calling it "a highly reactive species that can contribute to the formation of secondary organic aerosols". In fact, at atmospheric temperatures CH₃ can only rapidly recombine with other radicals but is virtually unreactive with closed shell species.

Using MD simulations with ab initio (DFT) potentials is fancy but the results are only as good as the underlying potentials. DFT methods do not generally provide chemical accuracy comparable with, e.g. coupled cluster theory. I wonder if the less than spectacular agreement of the measured and calculated photoelectron spectra (Fig. 3) is due to the deficiency of the DFT energetics. Also, in my opinion, more accurate estimates of the lifetime of the CH₃CO⁻ anion can be obtained using statistical (RRKM) calculations of its decomposition rate constant using more accurate (CCSD(T)-like) energies of the pertinent local minima and transition states, instead of using DFT-AIMD. In any case, validation of the chosen density functional vs. more accurate electronic structure methods is required before the results of the AIMD calculations can be trusted.

We thank the Reviewer for their suggestions and critical assessment of our work.

We start by addressing the Reviewer's comment on our calculations. We would like to stress that the calculated photoabsorption cross-sections obtained with the nuclear ensemble approach (NEA) are in excellent agreement with the experimental photoelectron spectra for pyruvate anion and CH_3^- . The additional calculated spectrum for the acetyl anion (dashed line in Fig. 3b) does not correspond to any observed photoionization signal in this experiment and was included to highlight this. Our calculated photoelectron spectra of the acetyl and methide anions can be compared to their reference *experimental* spectrum (solid lines below, with computed as dashed), highlighting the agreement and the identification of the fragment observed in our experiment with the methide anion:

Importantly, it is not only the energy of the band that is well reproduced for the photoelectron spectrum of the methide anion by our calculation, but also its overall shape. We believe that such excellent agreements between theory and experiment, and for the shape and energetic of the spectra, constitutes a strong validation of the theoretical protocol employed in this work.

For completeness, we have also performed the benchmark of our protocol proposed by the Reviewer, comparing the electron binding energy obtained with our (U)DFT/ ω B97X-D/aug-cc-pVDZ results with (U)CCSD(T)-F12/aug-cc-pVTZ-F12 for the three molecules of interest in our work (for each molecule, we employed the ground-state geometry obtained with DFT/ ω B97X-D/aug-cc-pVDZ). The results presented in the Table below shows the excellent agreement of our DFT binding energies with that predicted by CCSD(T)-F12.

	electron binding energy (eV)		
	Methide anion	Acetyl anion	Pyruvate anion
(U)DFT/ ω B97X-D aug-cc-pVDZ	0.381	0.606	3.786
(U)CCSD(T)-F12 aug-cc-pVTZ	0.395	0.665	3.906

Taken together, all these results strongly validate our theoretical methodology for the quantities compared with the experiment and leading to the unambiguous identification of the methide anion. We have included this additional information in the main text and the SI:

- Page 5: *To support this assignment, we calculated the photoelectron spectra of the pyruvate anion using the nuclear ensemble approach (NEA) with (U)DFT/ ω B97X-D/aug-cc-pVDZ (see discussion below and further information on the calculations in the Methods section).*

- Page 11 (Methods): *For each geometry, the vertical ionization energy to D_0 was calculated by a*

difference of electronic energy using DFT/ ω B97X-D/aug-cc-pVDZ for S_0 and UDFT/ ω B97X-D/aug-cc-pVDZ for D_0 , and the corresponding intensity was approximated by using the norm of the Dyson orbital. The electron binding energy obtained with this level of theory for each molecule at its ground-state geometry (obtained with DFT/ ω B97X-D/aug-cc-pVDZ) is in excellent agreement with that obtained with CCSD(T)-F12 (Table 1). The CCSD(T)-F12 calculations were conducted with Molpro 2012.

Table 1: Electron binding energy calculated with (U)DFT and (U)CCSD(T)-F12 for the optimised ground-state geometry of each molecule obtained with DFT/ ω B97X-D/aug-cc-pVDZ.

	electron binding energy (eV)		
	Methide anion	Acetyl anion	Pyruvate anion
(U)DFT/ ω B97X-D aug-cc-pVDZ	0.381	0.606	3.786
(U)CCSD(T)-F12 aug-cc-pVTZ	0.395	0.665	3.906

- We have included the figure comparing the experimental and computed spectra shown above as Supplementary Figure 3

The comment made by the Reviewer on our ab initio molecular dynamics (AIMD) made us realise that we should clarify further the use of these dynamics. These AIMD (described as 'exploratory' in our manuscript) have been used *solely* to understand which molecules could give the signal at low binding energies. They are not aimed to deduce any lifetime, as inferred by the Reviewer. The only information extracted from our AIMD is the suggestion that the acetyl anion could readily dissociate into CO and CH_3^- at high internal energy. This then led us to compute the photoelectron spectrum for CH_3^- which verified the origin of the experimental feature (Fig. 3 in our manuscript and further validation above). We also note that we have now calculated the dissociation energy (D_e) of acetyl anion to CO and CH_3^- and obtained a value of 113.4 kJ mol^{-1} with our DFT formalism and 93.1 kJ mol^{-1} with CCSD(T)-F12/aug-cc-pVTZ-F12. Both confirm that our exploratory AIMD calculations captured the view that the acetyl can dissociate.

To determine any kinetic information on this photodynamics process, one would need to perform excited-state dynamics to account accurately for the distribution of internal energy into the acetyl anion (see for example Ref. 37 cited in the manuscript about athermal dynamics). This is because both an AIMD or any RRKM calculations - as suggested by the Reviewer - would not be able to capture the non-statistical distribution of internal energy into the photoexcited pyruvate anion leading to the final formation of CO_2 , CO and CH_3^- . We have specifically avoided discussing the details of the dynamics as we do not have sufficient information on this at present.

We have clarified our use of ab initio molecular dynamics in the text.

- Page 8: *Specifically, during the AIMD conducted in the NVE ensemble, the ground-state acetyl anion appeared to be unstable with respect to CO loss already at an average temperature of 1400 K. Based on this observation, we calculated the photoelectron spectrum of CH_3^- and compared it with the experimental spectrum at $h\nu = 3.5$ eV (Fig. 3b).*

- Page 8: *In our experiment, we do not observe the acetyl anion. This is likely because CH_3CO^- is formed with sufficient internal energy to lose CO and is therefore expected to act as a short-lived intermediate ($< ns$) en route to forming the methide anion.*

- Page 13: *The dissociation energy (D_e) of acetyl anion leading to the formation of CO and CH_3^- is calculated to be 113.3 kJ mol^{-1} at the DFT/ ω B97X-D/aug-cc-pVDZ level of theory and 93.1 kJ mol^{-1} at the CCSD(T)-F12/aug-cc-pVTZ (using the same DFT optimised structures).*

We now address the comments of the Reviewer regarding the atmospheric implication of our work.

The Reviewer questions the importance of pyruvate as an isolated species in the atmosphere. As cited by the Reviewer, we recognized this fact in our manuscript. However, pyruvate is far more abundant in

the presence of water molecules (droplets, microsolvation, ice surfaces), as noted (and supported through Refs 17 and 18). To place this in perspective, for a sea-spray droplet (pH 8.1), there will be 4×10^5 anions for every pyruvic acid molecule (and $>2 \times 10^7$ at the surface)! Despite this abundance of pyruvate anion, it is surprising to realise that no information is available on its photochemistry, even in the gas phase. A first step in understanding the complex photolysis of pyruvate is therefore to investigate its photodecomposition as an isolated anion and understand which products are formed. We agree that our work highlights the photodissociation pathways that will take place in gas phase, but this raises many open (and potentially important) questions regarding their prevalence in more complex environment.

To address the comment of the Reviewer about the lack of kinetic data in our work, we would like to clarify that our main message is that the photodecomposition of pyruvate is likely to produce CO, CO₂, CH₃⁻, CH₃, and an electron. The possible presence of CH₃⁻ as a photoproduct of the conjugate base of pyruvic acid is an unprecedented measurement and these species have not been considered to date in Earth's atmosphere. To the best of our knowledge, the methide anion has been discussed in the context of Titan's atmosphere only (see Refs. 32 and 40 in the revised version of the manuscript). More generally, the presence of unexpected photoproducts has been detected by employing a novel set of spectroscopic and theoretical tools for the field of atmospheric chemistry. Being able to study directly transient species with new spectroscopic tools has had profound impacts in atmospheric chemistry, as has been demonstrated by, for example, the elegant laser-induced fluorescence experiments on Criegee's intermediates.

Finally, the Reviewer questions the relevance of CH₃ radical. its reaction with O₂ forms the methylperoxy radical (see Ref. 43). While the main source of CH₃ radical is the oxidation of methane by an OH radical, our findings show that a local formation of CH₃ (and possibly methylperoxide radical) can be expected where pyruvate is present. This would strike us as an important consideration for atmospheric chemists and modelers, given the importance of pyruvic acid in aqueous photochemistry.

We have modified the text in the following way to account for the Reviewer's comments on the atmospheric implication of our work. We further stress the production of CH₃⁻ and its decomposition into CH₃ and a free electron. We also highlight the importance of considering the production of CH₃ where pyruvate (and pyruvic acid) is present, in particular considering its reaction with O₂ to form a methylperoxy radical.

- Title: *Photochemistry of the pyruvate anion produces CO₂, CO, CH₃⁻, CH₃, and a low energy electron*

- Abstract: *The observation of the unusual methide anion formation and its subsequent decomposition into methyl radical and a free electron may hold important consequences for atmospheric chemistry.*

Page 1: *We demonstrate that the pyruvate anion not only experiences decarboxylation, but also a subsequent unimolecular decay to form CO and a methide anion, CH₃⁻. The methide anion further decomposes into CH₃ and a free electron.*

- Page 9: *Nevertheless, the study of the intrinsic dynamics offers important insight into the photo-induced decay pathways that are operable. The photochemical production of the methide anion – the simplest of carbanions – raises questions on its possible reactivity with other atmospheric compounds, which has not been considered previously, except in the atmosphere of Titan.^{32,40} The methide anion can also decay into CH₃ and a free electron. This low energy free (or partially solvated) electron can react with surrounding molecules.^{41, 42} The CH₃, which is normally formed from the reaction of methane with OH, reacts with O₂ to form methyl peroxide.⁴³ Hence, the previously overlooked pyruvate anion in gas phase has the ability to produce a range of exotic and reactive species in the atmosphere following photo-absorption.*

Report of Reviewer #2

Given the ubiquity of the pyruvic acid & pyruvate anion in solutions and at air/water interfaces, they play important roles in aerosol chemistry and atmospheric environments. Their photochemistry in actinic region is particularly relevant to their chemical transformations. Authors applied the state-of-the-art ion spectroscopy and advanced theoretical methods to carry out a comprehensive probe of the pyruvate anion in its isolated form. Authors excited the anion to its first excited state in the actinic regime, and watched the decay and dissociation channels via VMI photoelectron spectroscopy. Channels such as direct electron detachment, resonant processes that led loss of CO, CO₂ and producing CH₃-(methide) were detected, providing a rich and complex photochemistry for this important anion. Sophisticated theories were applied to support experimental observations. Overall this is a well executed work. The unraveled rich photochemistry should have important implications to understand the pyruvic acid transformation in the atmospheric chemistry setting. The discovery of producing methide and methyl radical is particularly worth noting. I would strongly recommend its publication.

We thank the Reviewer for their very positive feedback.

I only have a couple of technique questions:

what is the temperature used in simulating direct detachment channel for CH₃COCOO⁻ (Fig 3 a) via NEA approach?

All the simulations employing the NEA are done at OK, that is for a molecule in its ground vibrational state.

- Page 11: *For each molecule, a Wigner distribution for uncoupled harmonic oscillators was constructed from the DFT ground-state geometry and corresponding vibrational frequencies, from which 500 geometries were randomly sampled assuming that the molecule is in its ground vibrational state (OK).*

The weak spectral signature for CH₃⁻ is proposed via two-photon process. Was there any photon-flux dependent study conducted?

Yes, we did do what the Reviewer has suggested, and the results are now presented in the supplementary information. We prefer not to discuss it in the main manuscript simply because we feel it distracts from the flow and requires some space to explain the results. In essence, we observe that the CH₃⁻ signal scales linearly with photon flux. This may at first glance appear to contradict a 2-photon process (might expect quadratic behaviour), but the two separate photon absorptions have different cross sections (one for excitation of pyruvate and one for detachment from CH₃⁻). If either of these is significantly larger than the other – which is a reasonable expectation given their differing nature – then the one with the larger cross section will dominate and the photon flux dependence will appear as a one-photon process. We have added the following comment in the manuscript and discussed the details in the SI.

- Page 9: *The photon flux dependence of the CH₃⁻ peak is considered in Supplementary Data 1.*

Typo: top of page 4, Figure 1 - Figure 2

Corrected, thanks for spotting this typographical error.

Report of Reviewer #3

The communication describes a joint experimental and theoretical study of the title molecule, providing new information on its possible photodissociation pathways. How pyruvate decomposes plays a potential role in the cycles of atmospheric chemistry, and the fact that it may produce methide radicals is

a novel, potentially important, observation. The paper also shows novelty in the good agreement between experiment and theory, supporting the role calculations have to play in unravelling the signals from time-resolved spectroscopy and demonstrating the accuracy of present state-of-the-art calculations.

Both of these novelties are of significance for the field of chemical dynamics, and the decomposition pathway of pyruvate of wider significance in physical and atmospheric chemistry. The paper is a sound and well argued piece of work, and as a result I support it's publication in Nature Communications. I have a few minor points that the authors should consider.

We thank the Reviewer for their very positive feedback.

1. on p4 top "Figure 1 presents..." should refer to Figure 2.

Corrected, thanks for spotting this typographical error.

2. on p5 it is mentioned that the ground-state structure of pyruvate has a perpendicular carboxylate group, whereas pyruvic acid is planar. This is surprising as the electronic structure would not be expected to be very different so perhaps the authors could give a brief explanation as to why this is.

The reason for this twist of the carboxylate moiety is the repulsion between this negatively charged part of the molecule and the carbonyl group. The details of these findings are discussed in the references provided (Refs. 23-25) and we felt it sufficient to direct a reader to this literature.

3. on p5 it is noted that the experiments show that the photoelectrons from pyruvate come from a non-bonding p-orbital. Does this tie in with the calculated electronic structure for the radical ground-state?

This statement is corroborated by the calculation. The figure below shows the Dyson orbital for the ionization to D_0 of pyruvate. The main contribution to this Dyson orbital are the oxygens non-bonding p orbitals.

We included this figure into the Supplementary Material.

4. on p6 when discussing the "thermionic emission" signal, it is stated that this "... necessitates the absorption of a photon and the formation of a ground state anion". Does thermionic emission really require the absorption of a photon - it can occur if a molecule is heated to a high enough temperature? And does it really require that the electron is lost from the ground-state? Can it not occur for an electronic excited state as long as there is enough time for loss of coherence?

The Reviewer is technically correct - in principle, if the molecule is sufficiently hot, it will undergo thermionic emission. However, in the present experiment, such heating can only be achieved by an external perturbation (i.e. absorption of a photon). We thermalize our ions in a room temperature ion trap so ions should be approximately at 300 K. In any case, if they were very hot and were undergoing thermionic emission, then they would not survive the several 10s μ s time-of-flight in the mass-spectrometer. Note, that we have also performed similar experiments on many other anions (some more weakly bound) and have never encountered such problems. As for the second point, again, the Reviewer is

technically correct. The electron may be emitted from an excited if that excited state is sufficiently long-lived.

We have tightened up the wording based on the Reviewer's comments:

- Page 6: *Such signals are characteristic of statistical (or thermionic) electron emission, typically from the ground electronic state of an anion, ...*

- Page 7: *Some low energy electron emission may also arise from excited states by fast autodetachment that competes with internal conversion or dissociation*

5. on p6 it is stated that "Enhanced thermionic emission was observed in the $h\nu = 3.5 - 3.7$ eV spectra,...". I do not see this in Figure 2. Furthermore, what is the relevance of the excited-state that may lie at 3.6eV? May this provide a resonant state that results in thermionic emission (which contradicts the statement discussed above in Q4). Or is this the pathway to fragmentation of the excited anion?

We stated in the manuscript that "As each spectrum has been normalised to its maximum signal, comparison of photoelectron intensities across different photon energies should be done with caution and are not representative of excitation or detachment cross-sections." It is difficult to obtain the relative cross-sections in our experiments especially because of the near degeneracy of the excited state and the opening of the direct detachment continuum. We have therefore simply made the qualitative observation that we observe enhanced thermionic emission in the range noted. This can to some extent be appreciated from the signal to noise in these spectra.

We have added the comment "evidenced in Figure 2 by the enhanced signal-to-noise ratio in these photoelectron spectra," in the manuscript to support this observation.

The excited state at ~ 3.6 eV is the doorway to the formation of the photoproducts. It can in principle also contribute to the low-energy (thermionic) emission observed and is not in contradiction to the comments in Q4. If the excited state decays to the ground state of pyruvate, it will produce thermionic electrons. If this state goes on to produce photoproducts (as we observe), it will produce thermionic electrons. So, the argument of thermionic emission is fully consistent with our observations, even if we are not entirely certain on the details of the decay mechanism (and we specifically avoid a discussion of this because we are uncertain). Note that we stress on pg 6 that "But, regardless of the exact mechanism that leads to the feature peaking at $eKE = 0$ eV, its presence necessitates the absorption of a photon..." and we stand by this statement (see response to point 4).

- Page 7: *evidenced in Figure 2 by the enhanced signal-to-noise ratio in these photoelectron spectra.*

6. on p7 "a very weak feature at high eKE ..." is mentioned and then discussed. This is not visible in the curves of Figure 2 and this should be made clear.

We appreciate the Reviewer's comment. We considered that inclusion of the relative axes in Figure 3 would be sufficiently clear, but now appreciate that is probably not the case. Instead, we have now included a representative photoelectron spectrum as Figure 2(b) to emphasise this observation.

Figure 2: Photoelectron spectra following irradiation of the pyruvate anion using ns laser pulses with photon energy $h\nu$. (a) Spectra at a range of $h\nu$, normalised to their maximum intensity and offset for clarity. (b) Spectrum at $h\nu = 3.5$ eV in which the signal at $eKE > 2$ eV has been amplified to highlight a feature at high energy. Inset is a photoelectron image in which the centre has been saturated to highlight this high energy (large radius) feature. The polarisation of the light, ϵ , is indicated by the double arrow.

We discuss Figure 2b in the text:

- Page 7: In addition to the two clear electron emission features, a very weak feature at high eKE (low binding energy) appeared in all acquired photoelectron images up to $h\nu = 4.1$ eV, but with greatest intensity in the $h\nu = 3.5 - 3.6$ eV range. An example spectrum is shown in Figure 2(b) at $h\nu = 3.5$ eV demonstrating that this high kinetic energy signal has $< 1\%$ the intensity of the corresponding thermionic emission feature. In Figure 3(b), this feature is replotted in terms of electron binding energy.

- Page 9: Furthermore, the anisotropy of the feature *taken from the photoelectron image inset in Figure 2(b) was determined to be $\beta_2 \approx +0.4$, which agrees well with their modified Wigner-Bethe-Cooper-Zare equation for CH_3^- .*

7. on p7 the various possible fragments are discussed and discarded. For completeness, what about the formation of the CO_2^- ion?

The anion of CO_2 has not been observed previously except as a long-lived excited resonance state. CO_2 has a negative electron affinity and therefore, we did not consider it as a feasible intermediate. Moreover, if the long-lived excited state anion was formed, then this would have been picked up in our photoelectron spectra, but we do not see it. Finally, the fact that we see CH_3^- necessitates that the excess electron remains on the acetyl fragment rather than the CO_2 .

REVIEWERS' COMMENTS

Reviewer #1 (Remarks to the Author):

The authors have properly addressed all comments of the reviewers and convinced me on the importance of this work for atmospheric chemistry. Therefore, I recommend publication of this manuscript in its present form.

Reviewer #2 (Remarks to the Author):

In the revision, the authors have adequately addressed all my comments and suggestions in the original round of review. I am satisfied in particular their response regarding to the photon-flux dependent experiments on the yield of CH₃⁻. They appeared to successfully respond to comments and suggestions from other two reviewers. Therefore in my opinion, this revised manuscript is ready for publication in Nature Communications.

Reviewer #3 (Remarks to the Author):

In my opinion the authors have responded to all the comments of the three referees and made a strong case for the consistency of their results and what they show. They also make a strong case for the importance of the pyruvate anion photochemistry. I am certainly happy that my points have all been adequately addressed and the paper is stronger as a result. I recommend it for publication in Nature Communications.